# Carbon Amendments Shape the Bacterial Community Structure in Salinized Farmland Soil

Qisheng Han,[a,c] Yuanyuan Fu,[a,b] Rangjian Qiu,[d] Huifeng Ning,[a,c] Hao Liu,[a,c] Caixia Li,[a,c] Yang Gao[a,c]

[a]Farmland Irrigation Research Institute of Chinese Academy of Agricultural Sciences, Xinxiang, China
[b]College of Agriculture of Tarim University, Aral, China
[c]Farmland Irrigation Research Institute, CAAS/Key Laboratory of Crop Water Use and Regulation, Ministry of Agriculture, Xinxiang, China
[d]State Key Laboratory of Water Resources and Hydropower Engineering Science, Wuhan University, Wuhan, China

**ABSTRACT** Practical, effective, and economically feasible salt reclamation and amelioration methods are in great demand in arid and semiarid areas. Energy amendments may be more appropriate than alternatives for improving salinized farmland soil because of their effects on soil microbes. We investigated the effects of biochar (Carbon) addition and desulfurization (noncarbon) on the soil bacterial community associated with *Zea mays* seedlings. *Proteobacteria*, *Firmicutes*, and *Actinobacteriota* were the dominant soil bacterial phyla. Biochar significantly increased soil bacterial biodiversity but desulfurization did not. The application of both amendments stimulated a soil bacterial community shift, and biochar amendments relieved selection pressure and increased the stochasticity of community assembly of bacterial communities. We concluded that biochar amendment can improve plant salt resistance by increasing the abundance of bacteria associated with photosynthetic processes and alter bacterial species involved in carbon cycle functions to reduce the toxicity of soil salinity to plants.

**IMPORTANCE** Farmland application of soil amendments is a usual method to mitigate soil salinization. Most studies have concluded that soil properties can be improved by soil amendment, which indirectly affects the soil microbial community structures. In this study, we applied carbon and noncarbon soil amendments and analyzed the differences between them on the soil microbial community. We found that carbon soil amendment distinctly altered the soil microbial community. This finding provides key theoretical and technical support for using soil amendments in the future.

**KEYWORDS** carbon amendments, bacterial community and diversity, salinized farmland soil, stochastic or deterministic

Soil salinization affects a large land area globally (1) and has become a dominant cause of land degradation and crop yield reduction (2). A high salt concentration, high sodium cation ($Na^+$) concentration, and high pH lead to soil salinization intensified by $CO_3^{-2}$ concentration in soil (3). In arid and semiarid areas, low precipitation, high temperature, increased evaporation, and suboptimal irrigation scheduling have further exacerbated the situation (4, 5). For these reasons, salt accumulation in Xinjiang Province, the largest arid and semiarid region in China, increased by 40% from 1983 to 2005. Over one-third of all farmland is affected by irrigation-induced soil salinization (6). To maintain the sustainable development of agriculture in such areas, it is important to mitigate the effects of soil salinization.

Many studies have been conducted on saline soil remediation using physical, biological, and chemical methods (7). To date, leaching remains the most common and effective method to flush large amounts of salt out of the root zone by applying higher-quality water (8). But this strategy may be unsustainable in arid areas due to

Address correspondence to Yang Gao, gaoyang@caas.cn, or Qisheng Han, hanqisheng@caas.cn.

The authors declare no conflict of interest.

fresh water shortages. Hence, practical and economically feasible salt reclamation and amelioration methods are in great demand in both arid and semiarid areas (9).

Desulfurization gypsum is a common amendment for sodic soil reclamation because of its moderate solubility, ability to replace $Na^+$ at exchange sites with calcium ions ($Ca^{2+}$), low cost, and widespread availability (10). Values of soil electrical conductivity of the saturation extract (EC), pH, and sodium adsorption ratio (SAR) have significantly decreased after its addition (11). Sulfuric acid and elemental sulfur can also be used as alternatives to gypsum (12).

Recently, biochar has been considered as an alternative to reclaim degraded soil (13). Unlike the desulfurization gypsum, biochar is a carbon source amendment. As such, it is more likely to directly affect the soil bacterial community (14) and improve soil aggregate stability and hydraulic conductivity (15, 16). Biochar addition also reduced plant sodium uptake by transient $Na^+$ binding due to its high adsorption capacity, decreasing osmotic stress by enhancing soil moisture content and releasing mineral nutrients (particularly $K^+$, $Ca^{2+}$, and $Mg^{2+}$) into the soil solution (17). However, most studies have focused on the effects of soil amendment on improving soil physicochemical properties and plant-growing conditions; there have been few incubation studies on how types of amendments alter soil microbial communities.

Soil bacteria maintain mutualistic interactions with plant roots that enable plants to grow and tolerate abiotic stresses, including drought, salinity, heavy metal contamination, and pathogens (18). The extant literature stresses the role of soil microbes under salt stress, but how soil amendments to manage salinity have changed the structure of the soil microbiome has received little attention and needs further investigation. Therefore, the objectives of this study were (i) to understand the structure of the soil bacterial communities with different amendments, (ii) to investigate how different soil amendments shape the soil bacterial community, and (iii) to assess whether the change of soil bacterial community could affect the degree of host plants' salt resistance.

## RESULTS

**Differences in physiological indices among treatments.** The effects of DA (475 g soil plus 25 g desulfurized gypsum addition per pot) and BA (485 g soil plus 15 g biochar addition per pot) treatments on physiological traits differed. The BA treatment significantly improved net photosynthesis (Pn), stomatal conductance (Gs), and transpiration rate (Tr) values relative to the CK (500 g soil per pot) treatment, while no significant difference was found between the CK and DA treatments (Fig. 1a, b, and d). Oddly, there was no significant difference between the CK and DBA (460 g soil plus 15 g biochar addition plus 25 g desulfurized gypsum addition per pot) treatments for Pn, Gs, and Tr values. Intercellular carbon dioxide concentration (Ci) values were significantly higher for DBA than CK (Fig. 1c). The result clearly showed that soil water potential (Swp) values could be reduced by both soil amendments, but only the BA treatment exhibited a significant difference. Similarly, there was no change in Swp values relative to controls when two soil amendments were added simultaneously (Fig. 1e).

**Bacterial community composition.** A total of 2,104,510 reads were obtained after quality control and rarefaction and were clustered into 1,696 operational taxonomic units (OTUs) at 97% similarity. The species accumulation curve suggested that the sequencing effort recovered most local species' diversity (see Fig. S1 in the supplemental material). The dominant bacterial clades (top 9, phyla) in samples were *Proteobacteria* (48.2%), *Firmicutes* (23.3%), *Actinobacteriota* (11.0%), *Bacteroidota* (8.3%), *Gemmatimonadota* (3.4%), *Cyanobacteria* (2.4%), *Patescibacteria* (0.9%), *Verrucomicrobiota* (0.9%), and *Bdellovibrionota* (0.5%). At the family level, the DA treatment showed no significant changes in the relative abundance of dominant families compared to the CK treatment. The relative abundance of *Oxalobacteraceae* significantly increased in the BA treatment compared to CK, but we found no statistical difference with the DBA treatment. The DBA treatment significantly increased the relative abundance of *Pseudomonadaceae* and decreased the relative abundance of *Bacillaceae*. However, we did not find similar results for the DA and BA treatments (Fig. 2a, Fig. S2a). When analyzing the soil type data, the most dominant bacterial abundance was

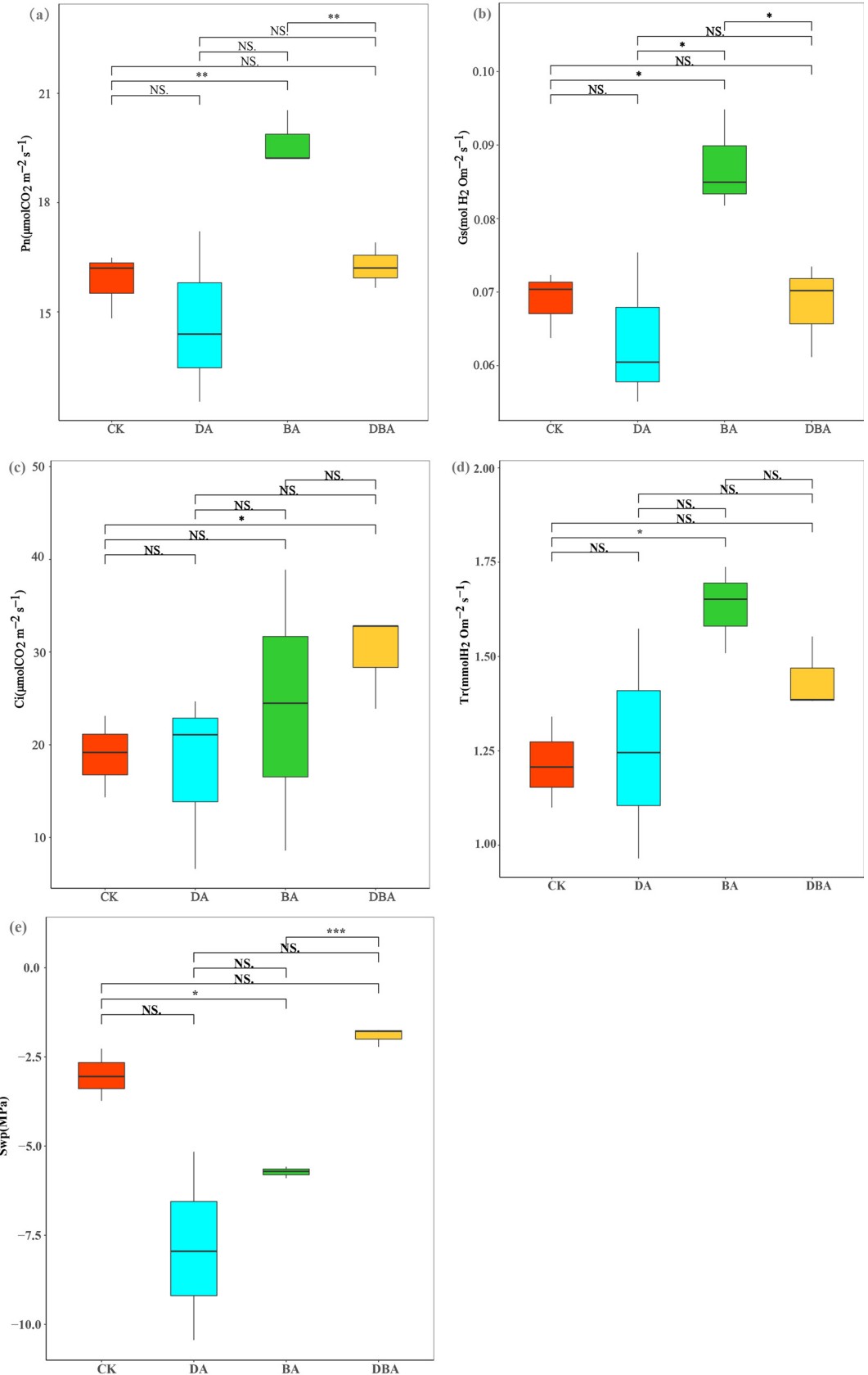

**FIG 1** Physiological indicators associated with leaf photosynthesis and water potential. (a to e) Box plots indicate the mean values of each treatment: (a) Pn, (b) Gs, (c) Ci, (d) Tr, and (e) Swp.

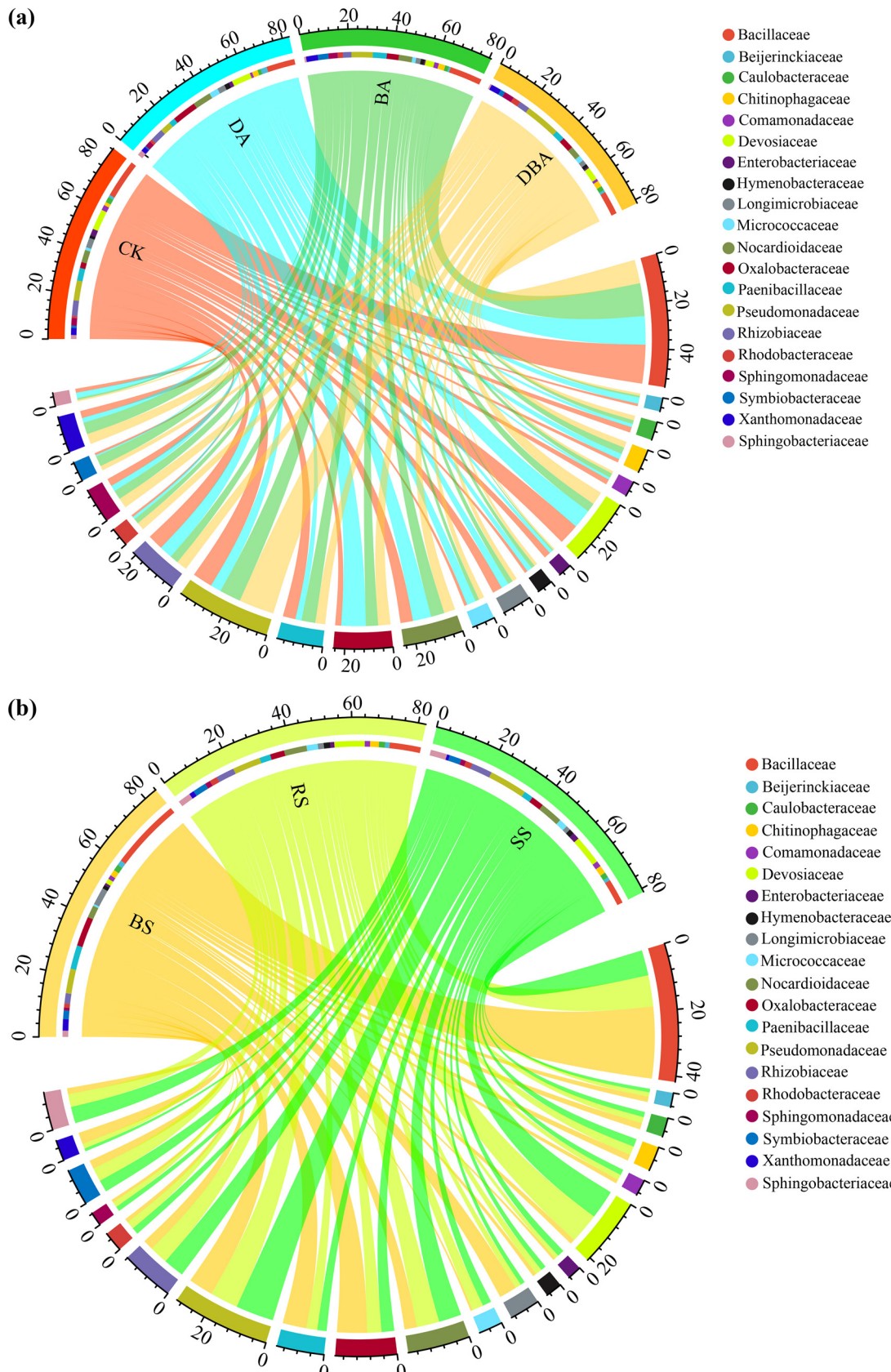

**FIG 2** Chord diagram based on the relative abundance of the top 20 members of the bacterial community at the family level. (a) Four treatments. (b) Three soil types. The chord diagram was visualized with the circlize package in R.

**TABLE 1** Biodiversity indices among treatments and soil types[a]

| Treatment or soil type | No. | Mean ± SE | | | | |
|---|---|---|---|---|---|---|
| | | Observed | ACE | Shannon | Simpson | Fisher |
| Treatments | | | | | | |
| CK | 9 | 732.1 ± 19.9 (b) | 913.8 ± 20.2 (c) | 4.66 ± 0.1 (b) | 0.975 ± 0.002 | 125.64 ± 4.19 (b) |
| DA | 9 | 736.7 ± 8.8 (b) | 944.5 ± 11.5bc | 4.68 ± 0.06 (b) | 0.974 ± 0.004 | 130.13 ± 1.17 (b) |
| BA | 9 | 823.4 ± 11.7 (a) | 1021.7 ± 16.4 (a) | 4.95 ± 0.04 (a) | 0.984 ± 0.001 | 146.27 ± 3.05 (a) |
| DBA | 9 | 781.4 ± 15.0 (ab) | 1005.9 ± 16.4 (ab) | 4.71 ± 0.05 (ab) | 0.974 ± 0.003 | 142.15 ± 2.8 (a) |
| Soil types | | | | | | |
| BS | 12 | 744.6 ± 17.8 | 934.4 ± 17.6 | 4.54 ± 0.07 (b) | 0.97 ± 0.003 (b) | 128.58 ± 3.56 (b) |
| RS | 12 | 789.8 ± 13.4 | 992.6 ± 15.8 | 4.89 ± 0.03 (a) | 0.982 ± 0.001 (a) | 142.21 ± 2.85 (a) |
| SS | 12 | 770.9 ± 15.6 | 987.4 ± 19.2 | 4.82 ± 0.04 (a) | 0.978 ± 0.002 (a) | 137.35 ± 3.11 (ab) |

[a]Different lowercase letters indicate statistically significant differences among treatments and soil types (one-way ANOVA and Tukey's *post hoc* comparison, $P < 0.05$).

not significantly different between rhizosphere soil (RS) and rhizoplane soil (SS) treatments. However, the abundance of *Devosiaceae* was higher, while the abundances of *Bacillaceae*, *Oxalobacteraceae*, and *Paenibacillaceae* were lower in the RS and SS samples than in bulk soil (BS) samples (Fig. 2b, Fig. S2b).

The BA treatment significantly increased the soil bacterial Chao1 and ACE indices, while both exogenous additions increased the Simpson and Shannon indexes (Table 1). Measures of soil type diversity revealed that the BS treatment had lower $\alpha$-diversity than the SS and RS treatments, but only the Simpson and Shannon indexes were statistically significant. At the species level, the Venn diagram (Fig. 3a) showed that the OTUs of the four treatments overlapped and shared 932 common OTUs. Bulk soil, rhizosphere soil, and rhizoplane soil shared 1,157 common OTUs, and unique OTU numbers decreased from bulk soil to rhizoplane soil (Fig. 3b).

**Correlating bacterial community structure and physiological indexes.** Principal-coordinate analyses (PCoA) of weighted UniFrac distance matrices (Fig. 4a) were conducted to find differences in the bacterial community structure among samples. PCoA showed distinct separation of samples from different treatments and soil types (Adonis: $R^2 = 0.28$, $P = 0.001$). The composition of the bacterial community had shifted in the BA and DBA treatments, and the microbial community did not change in response to the DA treatment. Furthermore, *Sumerlaeota*, TX1A-33, *Myxococcota*, *Bdellovibrionota*, and *Acidobacteriota* were significantly positively correlated with Ci, Tr, Pn, and Gs, while *Deinococcota* and *Chloroflexi* showed significantly negative relationships with Tr, Pn, and Gs (Fig. 4b).

We used the random forest method to analyze the most important bacterial phylum for classifying the four treatments and three soil types. *Deinococcota*, *Myxococcota*, *Bdellovibrionota*, and *Acidobacteriota* were the most important phyla in distinguishing community differences across treatments with low relative abundances (Fig. 5a), while *Gemmatimonadota*, *Halanaerobiaeota*, *Firmicutes*, and *Bacteroidota* were the most important phyla with high relative abundances across soil types (Fig. 5b).

**Ecological processes governing community structure.** We determined the differences between soil types and environmental parameters among sampling sites using distance-based redundancy analysis (db-RDA) (Fig. 6a). The first two axes accounted for 41.7% of the total variance. The db-RDA showed that treatments and soil type factors were significantly correlated with the bacterial community. To further quantify the relative importance of deterministic versus stochastic processes in shaping the bacterial community structure, we calculated the modified stochasticity ratio (MST). These results indicate that stochastic processes could play more important roles in controlling community succession through biochar addition, while deterministic processes could be more important with desulfurized gypsum and CK treatments. Consequently, adding biochar could relieve selection pressure and drive the community's more stochastic states. In addition, our results showed that MST values decreased with increasing distance from the surface of roots and hence led to a more deterministic outcome (Fig. 6b).

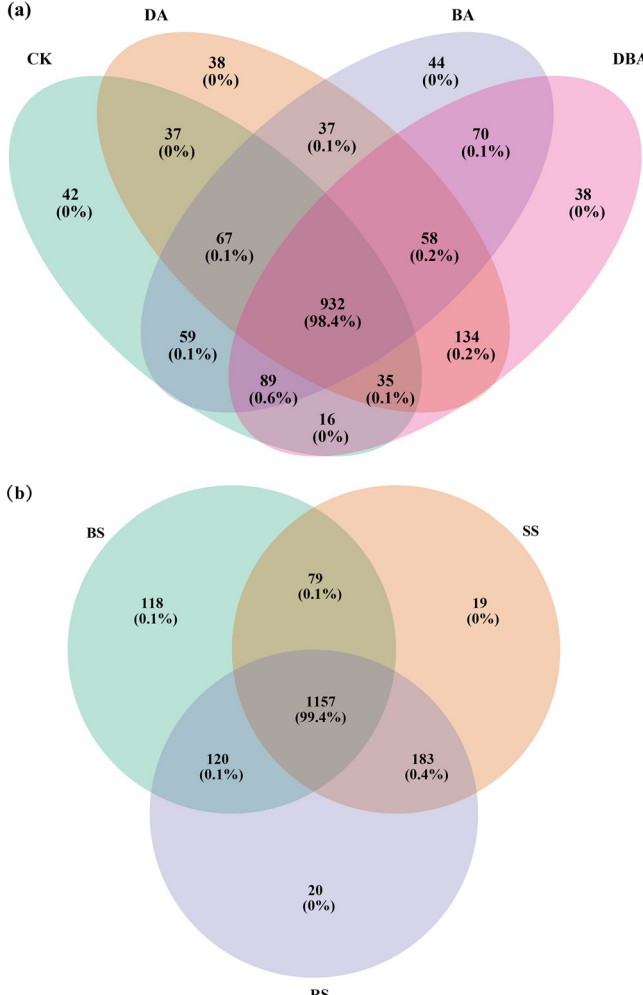

**FIG 3** Venn diagram of the identified bacterial species among different groups. (a) Four treatments. (b) Three soil types.

**Cooccurrence networks.** The cooccurrence patterns were analyzed to explore the potential roles of bacterial interactions. From the seven generated networks, we observed that the numbers of edges and total nodes were higher in the exogenous addition treatments than in the CK treatment and increased from soil to plant roots (Fig. 7, Table S3). For instance, we found that the number of nodes in the CK treatment was 267, while it was higher in the other treatments (DA, 316; BA, 298; and DBA, 272), indicating that the network of CK was less complex than those of soil amendments. The BS-type soil had the fewest nodes (297), compared with RS (348) and SS (365) soil types. In addition, the network of DA and BA treatments had a higher clustering coefficient and network density than CK, indicating that soil bacteria in similar niches were more closely linked than those in dissimilar niches after applying soil amendments. However, when the two soil amendments were added simultaneously, the clustering coefficient and network density were lower than CK. Changes were not limited to the numbers of nodes and edges but also occurred in the shift of dominant phylum. For example, *Actinobacteriota* was the dominant phylum in the CK network (Fig. 7a). However, the main phylum shifts to *Firmicutes* and *Proteobacteria* in the DA and BA treatments (Fig. 7b to d), respectively. Indeed, we found no dominant species in the BS soil type, and *Proteobacteria* and *Bacteroidota* appeared as the dominant phyla in the RS and SS soil types, respectively (Fig. 7e to g).

Microbial putative functions were analyzed using the functional annotation of prokaryotic taxa (FAPROTAX) to evaluate the potential functional differences among

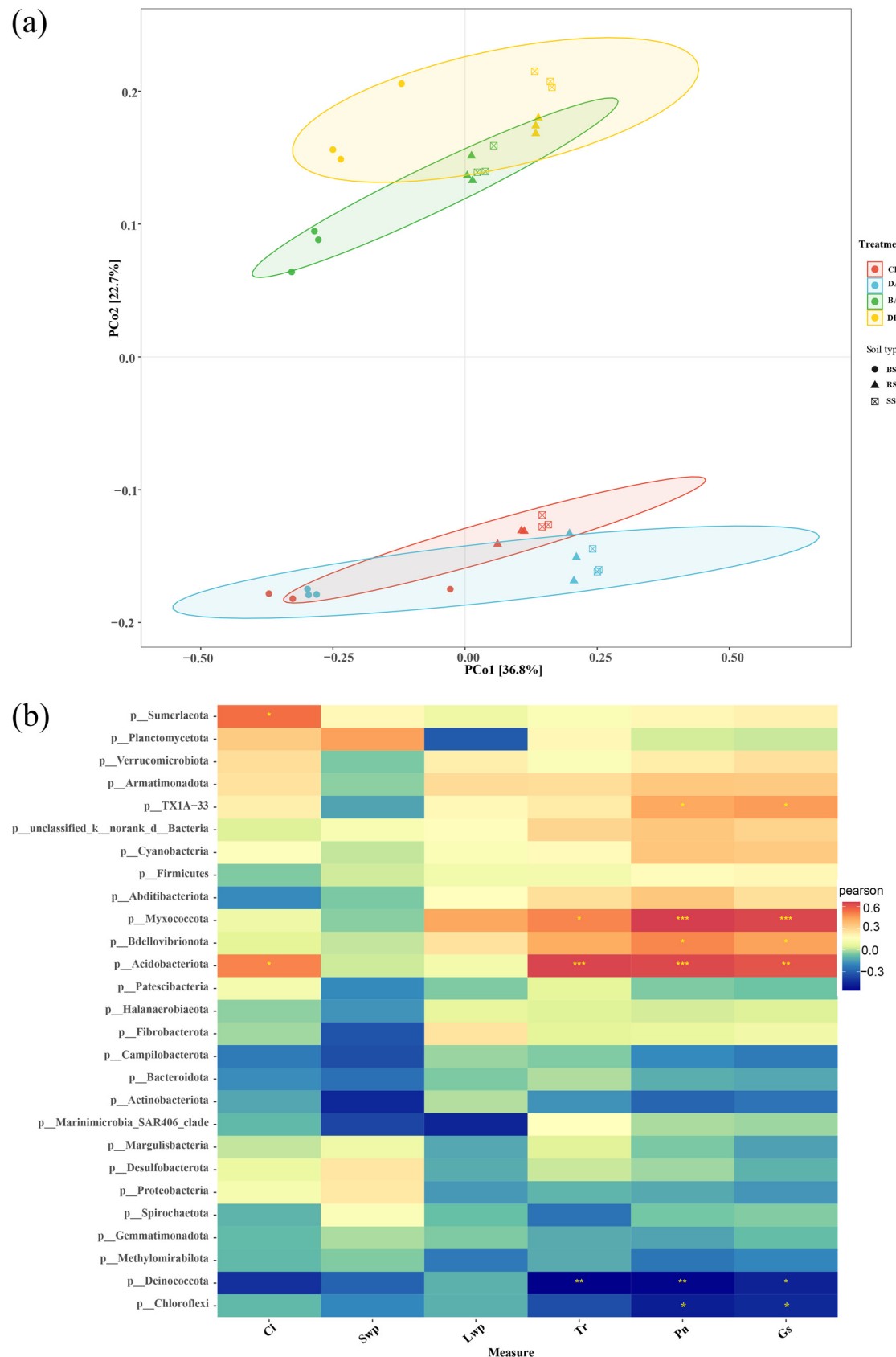

**FIG 4** (a) Bray-Curtis principal-coordinate analysis (PCoA) of samples based on UniFrac distance. (b) The heatmap of the correlations between physiological indexes and taxa at the phylum level determined by a Mantel test (***, FDR-adjusted $P < 0.001$; **, $P < 0.01$; *, $P < 0.05$).

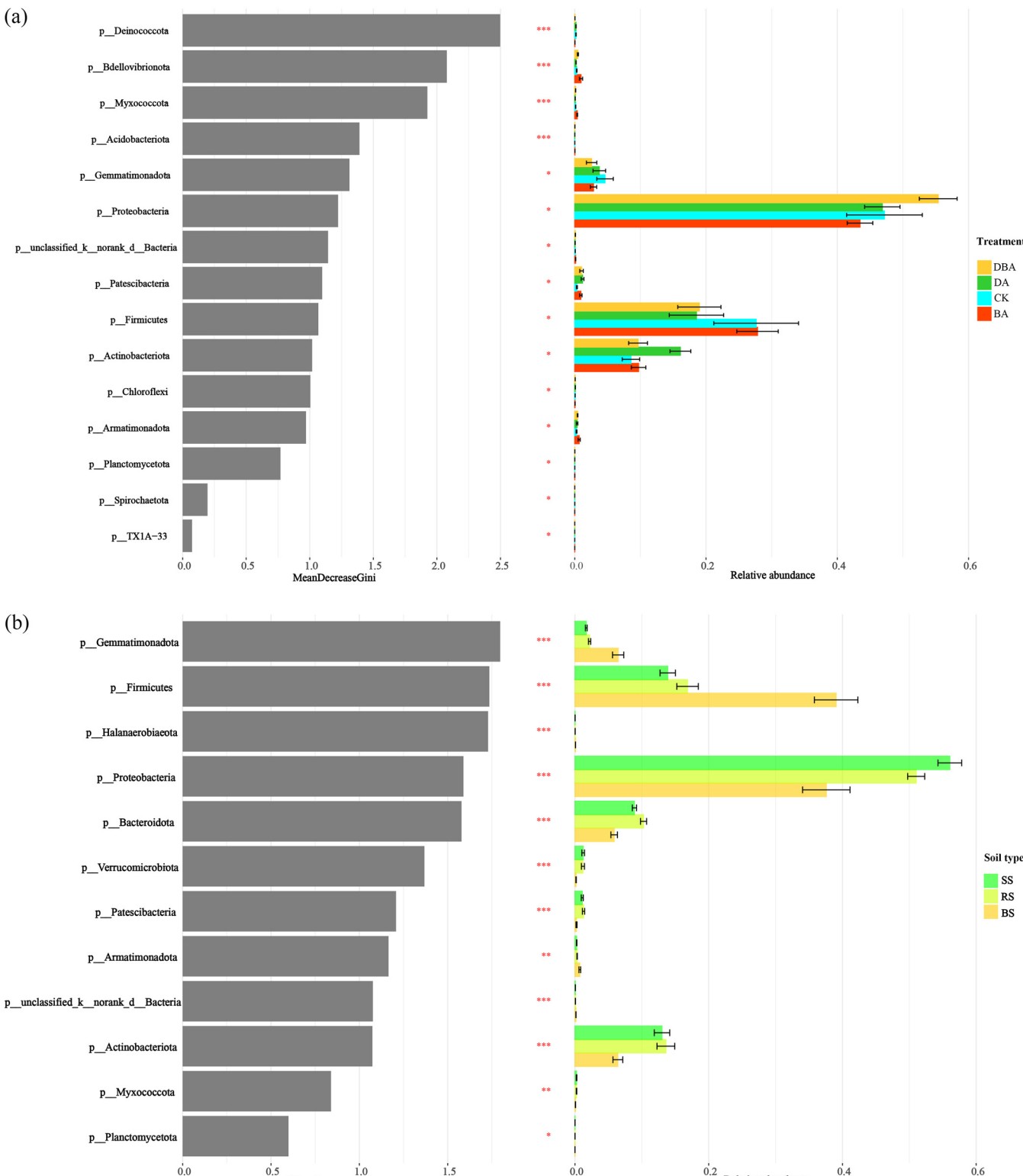

**FIG 5** The important bacteria at the phylum level and the histogram of the relative abundance of bacteria in different groups. (a) Four treatments. (b) Three type soils (***, $P < 0.001$; **, $P < 0.01$; *, $P < 0.05$).

samples. Energy source, C-cycle and N-cycle were the most abundant functions in all size factions (Fig. 8). The abundance of dominant bacterial species in BS soil was not affected by the amendments. The abundance of dominant bacterial species not assigned functions was higher than that of other taxa. Moreover, the dominant

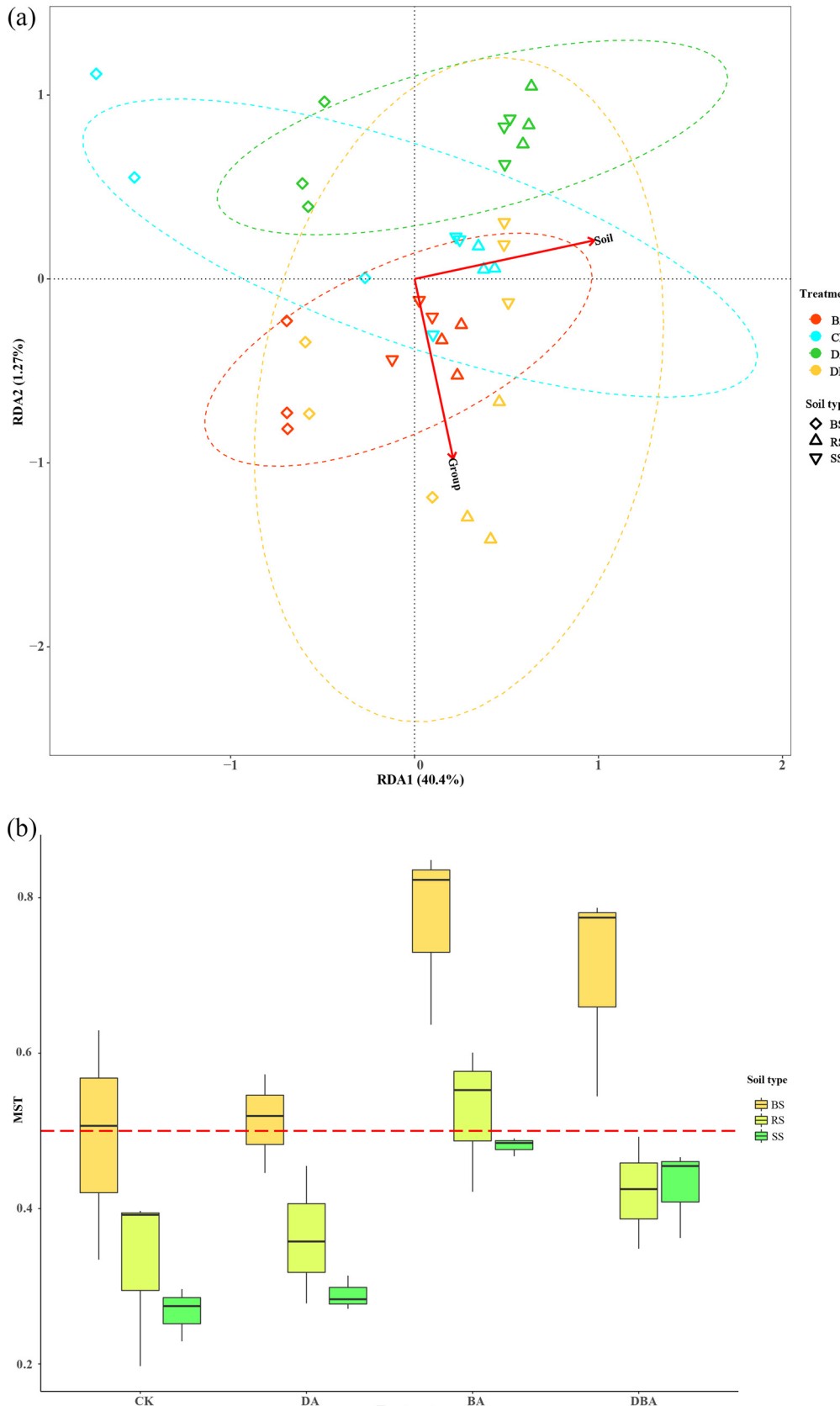

**FIG 6** (a) db-RDA analysis of bacterial communities at the phylum level in response to environmental parameters. (b) Null model analyses based on the modified stochasticity ratio (MST) of bacterial community structure.

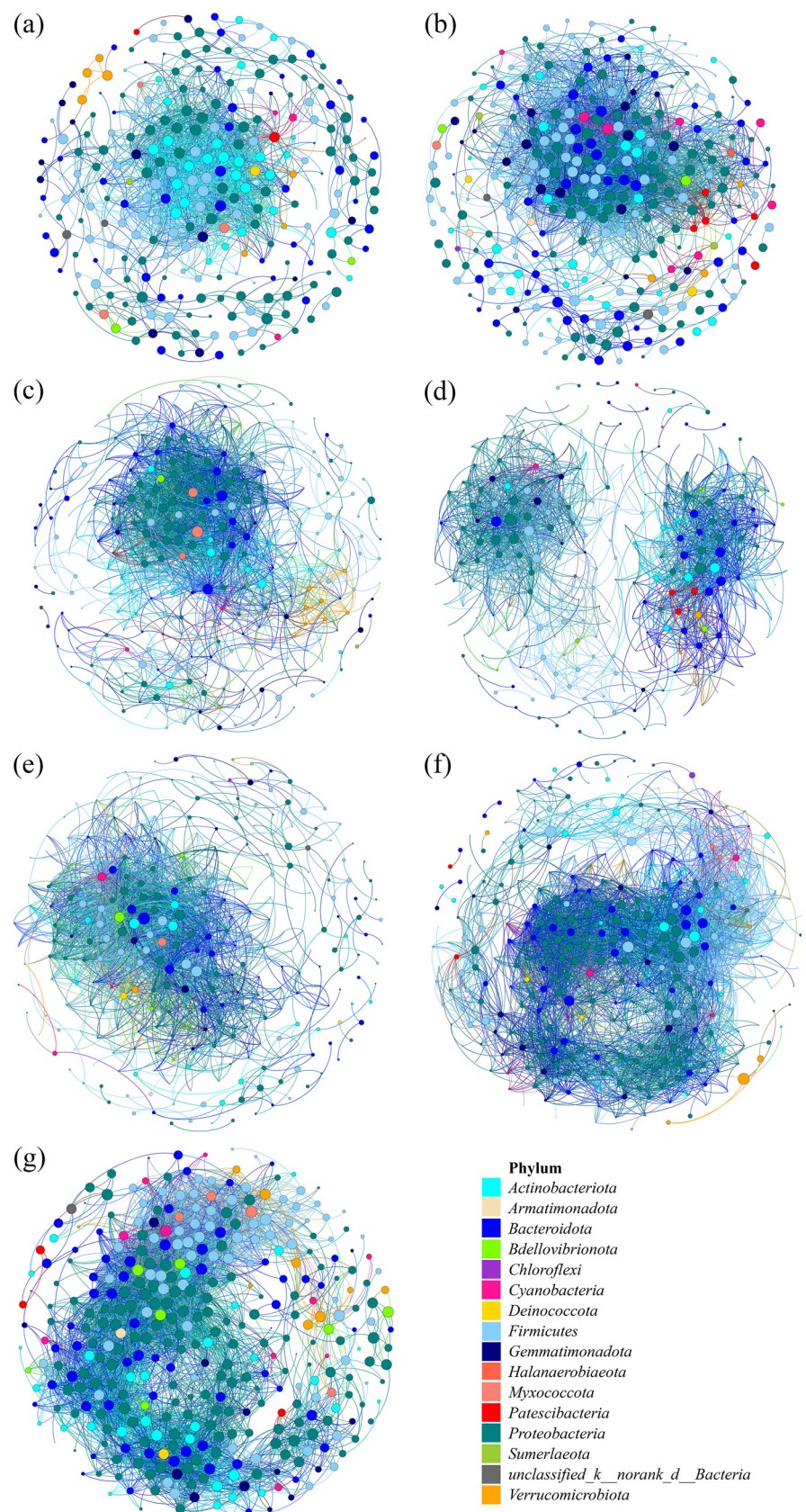

**Phylum**
- *Actinobacteriota*
- *Armatimonadota*
- *Bacteroidota*
- *Bdellovibrionota*
- *Chloroflexi*
- *Cyanobacteria*
- *Deinococcota*
- *Firmicutes*
- *Gemmatimonadota*
- *Halanaerobiaeota*
- *Myxococcota*
- *Patescibacteria*
- *Proteobacteria*
- *Sumerlaeota*
- *unclassified_k__norank_d__Bacteria*
- *Verrucomicrobiota*

**FIG 7** Cooccurrence networks of the bacterial community based on pairwise Spearman's correlations between OTUs. Each connection shown has a correlation coefficient of >0.7 and $P$ value of <0.01.

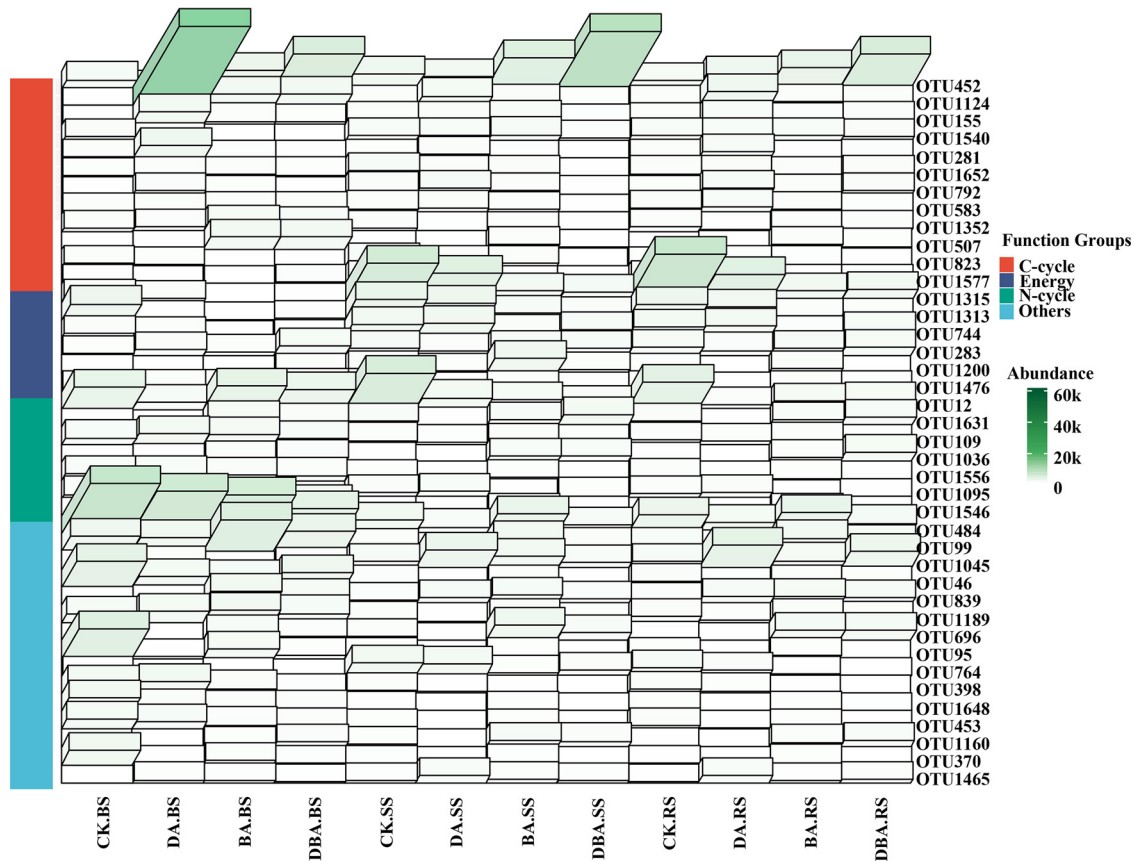

**FIG 8** 3D heatmap of the differences of the top 40 OTU abundances among all samples for functional groups of the FAPROTAX database (C-cycle, energy, N-cycle and others).

bacterial abundance was similar between CK and DA treatments in RS and SS soils, which showed high bacterial species abundance associated with energy and N-cycle. Adding biochar significantly increased the C-cycle properties of dominant bacterial OTUs in BS soil and SS soils.

## DISCUSSION

**Taxonomic composition of soil bacterial communities.** We performed extensive analyses of soil bacterial communities with different soil amendments with Illumina sequencing of the V3-V4 region of the 16S rRNA gene. Bacterial communities across all treatments were dominated by the phyla *Proteobacteria* (431 OTUs), *Firmicutes* (531 OTUs), and *Actinobacteriota* (118 OTUs), which was consistent with previous studies of the relative abundances of bacterial communities in agricultural soils in Xinjiang, China (19, 20). However, the relative abundance of *Proteobacteria* (48.2%) was found to be greater than in other studies (19, 20). It appears that *Proteobacteria* are often associated with plant rhizospheres and are favored by the presence of plants (21), especially in long-term cotton cropland (22). However, the greatest relative abundance of *Bacillaceae* at the family level does not belong to the *Proteobacteria*, which differs from a former study (23). A possible explanation was that the high soil pH (8.07, Table S1) in this study triggered the increasing abundance of *Bacillaceae* groups (24). In addition, the relative abundance of *Pseudomonadota* (*Oxalobacteraceae* and *Pseudomonadaceae*) significantly

**FIG 7** Legend (Continued)

The size of each node is proportional to the number of connections. The top panel shows the network of four treatments with OTUs colored by phylum: (a) CK, (b) DA, (c) BA, and (d) DBA. The bottom panel shows the network of three soil types with OTUs colored by phylum: (e) BS, (f) RS, and (g) SS.

increased and that of *Firmicutes* (*Bacillaceae*) decreased when biochar was added to soil. The higher relative abundance of *Pseudomonadota* was probably explained by the fact that *Pseudomonadota* are considered copiotrophic bacteria and flourish in soils with large amounts of available nutrients (25), and the *Firmicutes* were suppressed by the soil amendment providing carbon and other nutrients (26).

Bacterial diversity as well as clear changes in the bacterial community were previously shown to be linked to the soil amendments (27, 28). The data we obtained show that bacterial biodiversity was significantly improved with the addition of biochar but not with the other additions (Table 1). Bacterial diversity increased significantly in biochar-amended soil and was positively correlated with the addition ratio of biochar (14). Bacteria may adhere to biochar surfaces, rendering them less susceptible to leaching into soil, increasing bacterial abundance with biochar addition (13). Our experimental results demonstrate this effect (Table 1, Fig. 3a). Indeed, there was little difference in the biodiversity index of the three soil types. In the rhizosphere soil, the bacterial diversity was slightly higher than in the bulk soil and rhizoplane soil (Table 1), indicating that bacterial colonization of roots is not a passive process and that the host plants have the ability to select for certain microbial consortia (29).

Ecological succession and the balance between stochastic and deterministic processes are two major themes in microbial ecology (30). Ecological patterns are used to characterize measured and unmeasured abiotic variables that impose selection, and differences in the physical energy of geological depositional processes can result in between-formation environmental differences resulting in turnover due to selection (31). In our study, the soil bacterial community structure between the CK and DA treatments were similar, but they were clearly separated from the BA and DBA treatments (Fig. 4a). Correspondingly, the MST values of the bacterial community for the BA treatment were more than 0.5, which was higher than those of other treatments (Fig. 6b). This indicates that stochastic processes dominated the soil bacterial community assembly with biochar addition. Particularly in microbial ecology, diversification is often considered a component of stochastic processes. Ecological drift is defined as stochastic changes with respect to species identity in the relative abundances of different species within a community over time (32). The soil properties among treatments in this study are broadly consistent, so the bacterial community in the soil should have the same development process. However, soil organic carbon was the major driver of bacterial $\beta$-diversity, and bacteria were more influenced by undominated-species stochastic processes (33). Dini-Andreote et al. (30) also found that an increased resource supply can increase stochasticity under physicochemical conditions that do not impose strong selection. In addition, Luan et al. (34) found that the stochastic process of bacterial community assembly was influenced by soil carbon metabolism. Therefore, after adding the biochar, we suppose that the increasing carbon-related bacterial species would enhance the function and role of soil microbial communities to develop resistance to soil salinity stress.

Microbial communities with more complex cooccurrence networks were indicative of greater ecological balance and complexity, which was more resistant to environmental stresses than those with simpler networks (35). Wang et al. (36) found that the high-salinity soil (EC, >4) harbored less ecologically similar functional groups and was expressed primarily in the low bacterial network connectivity. Our results illustrated that the complexity of bacterial networks increased with the application of BA and DA (Fig. 7, Table S4), which means that the soil microbial communities were more resistant to salt stress than CK treatment. Interestingly, there was no synergistic effect on soil bacterial networks when the two amendments were added, and antagonism was even observed (Table S4). Liu et al. (37) also found that applying the soil amendments increased antagonisms between microbial taxa, especially bacteria.

However, many microbial types can survive for extended periods of time in unfavorable habitats by entering a dormant state (38). Dormant microorganisms generate a seed bank, which comprises individuals that are capable of being resuscitated following environmental

change (39). The soil amendments may act as a hook to stimulate select bacterial propagules in the soil seed bank. Five phyla (*Proteobacteria*, *Actinobacteria*, *Firmicutes*, *Acidobacteria*, and *Bacteroidetes*) contained 90% of the bacterial sequences in saline soils. Different soil amendments increased the interactions of different bacterial phyla. In our study, adding the biochar enhanced the interaction among the *Proteobacteria* phyla, while desulfurization affected the *Firmicutes* phyla. *Proteobacteria* and *Firmicutes* were reported as good indicators for reflecting changes of the main microbial groups and an important resource for exploring halophilic enzymes and metabolic pathways for pollutant remediation in saline soil (36, 40). This might be another reason that the stochastic processes became dominant in shaping the bacterial community assembly after adding the soil amendments. Nevertheless, the effect of biochar amendment on soil bacterial richness and diversity remains controversial (41, 42). It is well known that biochar amendment can shift the soil bacterial community structure (43). In particular, the bacteria in the rhizosphere soil showed clearer responses to biochar addition than the bacteria in the bulk soil (Table 1), and the proportional abundance of *Proteobacteria* increased significantly after biochar addition (43, 44). Consequently, *Proteobacteria* became the dominant phylum in the BA treatment.

Microbial community composition and plant productivity usually showed a strong link (45). Photosynthetic capacity is one feasible criterion for identifying plant salt tolerance based on its sensitivity to salt stress (46). However, we found that some bacterial phyla were significantly associated with seedling photosynthesis indicators (Fig. 4b). Importantly, these phyla were the most important bacteria accounting for the treatments' differences by the random forest analysis (Fig. 5a). In addition, following the biochar addition into the soil, the relative abundance of each bacterium positively correlated with photosynthesis indicators (*Myxococcota*, *Bdellovibrionota*, and *Acidobacteriota*) increased. The latest research showed that anoxygenic photosynthesis genes were detected in *Myxococcota* and *Acidobacteriota*, which play an important role in the absorption of light for photosynthesis (47). On the other hand, soil bacteria also contribute to many essential soil processes such as the carbon and nitrogen cycles (48), and plant growth and production may be regulated by soil carbon and nitrogen cycling (49). Moreover, we also found that the absolute abundance functional groups involved in the Carbon cycle accounted for much of the total sequences and changed with different treatments (Fig. 8). Specifically, the number sequences of OTU 452 assigned to the family *Pseudomonadaceae* increased with biochar addition. It has been found that *Pseudomonadaceae* were the initial colonizers after sterilization and reinoculation of an arid soil, and we attributed this to the enzymatic capacity of members of this family to initiate the degradation (50). They are also particularly well suited for plant root colonization, and many strains display plant growth-promoting and/or biocontrol activity against various plant pathogens (51). Therefore, we speculate that the biochar could enrich soil pioneer bacterial groups, which in turn would improve the soil environment and enhance seedling growth. Notably, the abundance of OTU 1315, belonging to the genus *Devosia*, decreased in rhizosphere and rhizoplane soil with the addition of biochar (Fig. 8). This is the same as the research result of Xu et al. (52). In addition, *Devosia* organisms are well known for their dominance in soil habitats contaminated with various toxins (53). This may be another proof that biochar can reduce the poisonous effect of salt stress on seedling roots.

## MATERIALS AND METHODS

**Plant materials, growth conditions, and treatments.** Experiments were carried out at the Key Laboratory of Crop Water Use and Regulation, Ministry of Agriculture and Rural Affairs, China, located in Xinxiang (35°9′N, 113°47′E; elevation, 74 m). The experimental soils were collected from farmland abandonment where maize was previously cultivated in the Aral District, Xinjiang Province, northern China. The soil type is classified as sandy loam, and the sample soil was air-dried, ground, and sifted through a 5-mm sieve before the experiment (Table S1). The experimental design included four treatments: 500 g soil per pot (CK), 475 g soil plus 25 g desulfurized gypsum addition per pot (DA), 485 g soil plus 15 g biochar addition per pot (BA), and 460 g soil plus 15 g biochar addition plus 25g desulfurized gypsum addition per pot (DBA). Six replicates were maintained per treatment. The biochar was derived from the carbonization of corn straw at 400°C, and the carbon content was within 53.1% to 57.1%. Other elemental analysis results are listed in Table S2. Two corn seeds (*Zea mays*, c.v. Chenghai 867) with plump and uniform size were sown at a 20-mm depth in each pot (10-cm diameter, 9-cm height). Pots were irrigated with 200 mL

distilled water prior to sowing and placed in an artificial climate chamber (25/20°C day/night temperature; photoperiod, 12 h light; humidity, 40% $\pm$ 5%). After germination, only a single seedling was retained per pot. Seedlings were irrigated on days 7 (150 mL), 9 (150 mL), and 13 (100 mL) with Hoagland nutrient solution (Table S3). The salt nutrient solutions were added to treatment pots on days 15 (100 mL) and 17 (50 mL) to create a high-salt environment (Table S3).

**Physiological trait analysis.** After 20 days of seedling growth (two-leaves stage), three pots were randomly selected in each treatment group to determine the photosynthesis and leaf water potential of seedlings and collect soil samples for microbial analysis. Measurements of photosynthesis were conducted using the youngest fully expanded main-stem leaf (the 1st leaf from the apex). Photosynthetic parameters of leaves, including net photosynthesis (Pn, $\mu$mol $CO_2$ m$^{-2}$ s$^{-1}$), stomatal conductance (Gs, mol $H_2O$ m$^{-2}$ s$^{-1}$), intercellular carbon dioxide concentration (Ci, $\mu$mol $CO_2$ mol$^{-1}$), transpiration rate (Tr, mmol $H_2O$m$^{-2}$ s$^{-1}$), were measured from 9:00 to 11:00 a.m. using a Li-6400 portable photosynthesis system (LI-COR, Inc., Lincoln, NE, USA). The leaf was placed under 1,000 $\mu$mol m$^{-2}$ s$^{-1}$-intensity light, and the ambient $CO_2$ concentration was approximately 400 $\mu$mol $CO_2$ mol$^{-1}$. Then, leaf water potential (Lwp, Mpa) was measured with a water potential meter (WP4C, Decagon Devices, USA). Seedling leaves were cut and placed inside the sample cup. The soil water potential (Swp, Mpa) was determined with a WP4C water potential meter (Meter, USA). All measurements were repeated three times.

**Sample collection of bulk soil, rhizosphere soil, and rhizoplane soil.** Soil samples were separated into three types: bulk soil (BS), rhizosphere soil (RS), and rhizoplane soil (SS). For bulk soil collection, samples were taken from approximately 5 cm below the soil surface. The rhizosphere soil and rhizoplane soil were gathered according to the method described in Edwards et al. (29).

**DNA extraction and sequencing.** For each treatment, triplicates of BS, RS, and SS samples were used for total DNA extraction. Microbial DNA was extracted using the E.Z.N.A soil DNA kit (Omega Bio-tek, Norcross, GA, USA), and DNA purification was determined using a NanoDrop 2000 UV-vis spectrophotometer (Thermo Scientific, Wilmington, NC, USA). DNA quality was checked by 1% agarose gel electrophoresis. The distinct regions of the 16S rRNA (V3-V4 hypervariable regions) were amplified using specific primers (338F: 5'-ACTCCTACGGGAGGCAGCAG-3'; 806R: 5'-GGACTACHVGGGTWTCTAAT-3') (GeneAmp 9700, ABI, USA). PCRs were performed using the following process: denaturation at 95°C for 3 min, 27 cycles of 30 s at 95°C, annealing at 55°C for 30s, elongation at 72°C for 45s, and extension at 72°C for 10 min. PCRs were performed in triplicate in 20-$\mu$L volumes which contained 4 $\mu$L of 5 $\times$ FastPfu buffer, 2 $\mu$L of 2.5 mM deoxynucleoside triphosphate (dNTP), 0.8 $\mu$L of each primer (5 $\mu$M), 0.4 $\mu$L of FastPfu polymerase, and 10 ng of template DNA. The PCR products were extracted from a 2% agarose gel and quantified using a QuantiFluor-ST system (Promega, USA).

Purified amplicons were pooled in equimolar concentrations for subsequent sequencing on a MiSeq platform (Illumina, San Diego, CA, USA) by Majorbio Bio-Pharm Technology Co. Ltd. (Shanghai, China). Raw data were deposited in the NCBI Sequence Read Archive (SRA) database (Accession Number SRP345918). Raw fastq files were demultiplexed, quality-filtered using Trimmomatic, and merged using FLASH. Operational taxonomic units (OTUs) were clustered with a 97% similarity cutoff using UPARSE (version 7.1; http://drive5.com/uparse/). Taxonomy assignment was performed using the RDP Classifier algorithm (http://rdp.cme.msu.edu/) against the Silva 138/16s_bacteria (SSU123).

**Statistical analysis.** Differences in photosynthesis and leaf water potential values among treatments were analyzed by using one-way analysis of variance (ANOVA) with R software (54). The rarefaction curves for each sample were calculated by using the iNEXT package in R (55). The OTU abundance at each taxonomic rank and the $\alpha$-diversity indices and $\beta$-diversity indices for each treatment were calculated using the functions cal_abund (), cal_alphadiv, and cal_betadiv, respectively, from the R package microeco (56). Principal coordinate analysis (PCoA) was used to visualize the dissimilarity with the cal_ordination function from the R package microeco. PERMANOVA was used to test the significance of distances among groups with the R package vegan (57), using the adonis function with 999 permutations. Correlations between physiological traits and taxa at the phylum level were analyzed using the false-discovery rate (FDR) in the R package microeco. The random forest method (58, 59) was used to identify biomarkers at the phylum level by using the trans_diff function in the R package microeco. MeanDecreaseGini was selected as the indicator value.

The relationship between bacterial communities and environmental parameters was explored using distance-based redundancy analysis (db-RDA) in the R package microeco. The relative abundance of bacteria at the phylum level and measured environmental parameters were used as species and environmental inputs, respectively. Deterministic and stochastic processes represent two complementary parts along a continuum of ecological forces shaping community structure (60). The deterministic process generally refers to nonrandom ecology processes, such as environmental filtering (e.g., temperature, pH) and biological interactions (e.g., competition, mutualisms), while the stochastic process is a diversity pattern indistinguishable from random chance, including random birth-death, random dispersal, and ecological drift (61). The modified normalized stochasticity ratio (MST) was determined to evaluate the relative importance of deterministic and stochastic processes to the treatment among bacterial communities. This metric estimates ecological stochasticity according to a null-model-based statistical framework as described in Guo et al. (62). Unlike the null-model-based framework mentioned above, MST reflects the contribution of stochastic processes based on relative differences between the observed situation and the null expectation, rather than the significance of the difference, and therefore can better quantitatively measure the stochasticity in assembly (63). The value of the MST index was developed with 50% as the boundary point to divide the deterministic dominance ($<$50%) and stochastic dominance ($>$50%) community assembly. The MST analysis was performed based on the taxonomic assignment by using the Bray-Curtis distance and was implemented in the R package NST (64, 65).

Cooccurrence networks were constructed using the R package microeco. We performed weighted correlation network analysis (WGCNA) to find modules of high correlation (66). Only OTUs with a proportion above 0.01% across all samples were retained, and pairwise Spearman's correlations between OTUs were calculated. The network was visualized in Gephi (https://gephi.org). Functional profiles of the top 40 bacterial OTU abundances were predicted using the FAPROTAX database (67), from which three functions relevant for this study were selected: C-cycle, energy, and N-cycle. The results were represented by a 3D-heatmap using the R package ComplexHeatmap (68).

**Data availability.** Raw data were deposited in the NCBI Sequence Read Archive (SRA) database (accession number SRP345918).

## SUPPLEMENTAL MATERIAL

Supplemental material is available online only.
**SUPPLEMENTAL FILE 1**, PDF file, 0.3 MB.

## ACKNOWLEDGMENTS

This work was supported by the Major Program of the National Natural Science Foundation of China (grant no. 51790530), the National Natural Science Foundation of China. (grant no. 51809269), the Co-ordination Foundation of the Chinese Academy of Agricultural Sciences. (no. FIRI 2022-02), and the National Cotton Industrial Technology System (CARS).

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
