## [Reviewer comments · Microbiology Spectrum]

Microbiology Spectrum

Carbon amendments shape the bacterial community structure in salinized farmland soil

Qisheng Han, Yuanyuan Fu, Rangjian Qiu, Huifeng Ning, Hao Liu, Caixia Li, and Yang Gao

Corresponding Author(s): Qisheng Han, Farmland Irrigation Research Institute

Review Timeline:

Submission Date:	March 23, 2022
Editorial Decision:	July 24, 2022
Revision Received:	September 29, 2022
Editorial Decision:	November 29, 2022
Revision Received:	December 2, 2022
Accepted:	December 15, 2022

Editor: Wen-Li Chen

Reviewer(s): The reviewers have opted to remain anonymous.

Transaction Report:

DOI: <https://doi.org/10.1128/spectrum.01012-22>

July 24, 2022

Dr. Qisheng Han
Farmland Irrigation Research Institute
Road Honglidadao 380
Xinxiang, Henan
China

Re: Spectrum01012-22 (Energy amendments much easier shape the bacterial community structure in salinized farmland soil)

Dear Dr. Qisheng Han:

Thank you for submitting your manuscript to Microbiology Spectrum. I can now inform you that the Editorial Board has evaluated the manuscript Spectrum01012-22: Energy amendments much easier shape the bacterial community structure in salinized farmland soil.

The manuscript has been reviewed carefully by two experts (comments given below) who suggest that it requires significant improvements in many aspects including language, presentation, data interpretation. More importantly the manuscript requires extensive changes in the language and presentation.

The Editor has advised that the manuscript will be reconsidered for publication after thorough revision.

Link Not Available

Sincerely,

Wen-Li Chen

Journals Department
Reviewer comments:

Reviewer #1 (Comments for the Author):

This paper reports on an experiment in which different laboratory soil mesocosms that used relatively saline soils to which biochar and gypsum additions were done to test what effects these substances might have on basic plant physiology parameters and changes in the soil microbiome. There is significant interest in the use of biochar as a soil amendment to improve soil health and potentially improve C-uptake and sequestration within soils; however experimental evidence for the role of biochar is generally lacking, thus this work has the potential to aid our understanding of its role in these processes.

The paper as written is difficult to follow. Principally, the authors refer to biochar as an 'energy substrate', but provide no evidence that biochar itself is used as a microbial energy source, either electron acceptor or donor, directly. The experimental system itself is a highly contrived single laboratory experiment, thus field relevance is harder to interpret, and experimental details are missing. The interpretation of the data is, in places, speculative to the point that basic ecological principles are difficult to follow, in part because this is entirely a 16S amplicon based community study. Some further specifics are given below.

Specific comments.

The title is uninformative as written since it's not clear what 'energy amendments' are.

I96, Unclear what is meant by 'pots were poured'? Does this mean 200ml of water added? What's the total volume of soil? What was the moisture content before and after addition.

I. 100. Biochar is simply described as maize, 400C, this is inadequate. Biochar itself is a complex compound, and the specific conditions under which it is produced can influence its properties, thus significantly more detail is needed here. Furthermore, the amounts of addition and % in soil are not given for any of the additions.

I104/5 What was the original salt concentration of these soils.

P12. It's not clear what 'modified stochastic ratio' is or means, nor it is explained in the text. My general interpretation of the results is that there is a modest statistically significant impact of the biochar treatment on the microbial community, while other treatments were not significant. Thus, it's possible that the biochar may have, to some extent, selected for a different community, which would be determined by some, not identified here, property that the microbes are responding to. It's not clear to me how this is 'stochastic'. Furthermore, without more experimental details of how amendment concentrations and how these effect total soil moisture and organic C contents, it's a bit difficult to fully interpret.

lower part pg 14 & pg15, A number of studies are cited, however it's unclear how this work directly links to these studies and adds something new.

P15 The last statement of the MS about effects of biochar is largely conjecture. This experiment did not directly test nutrient mobilization or microbial growth as a result of biochar amendment to lend direct support to these ideas, since the microbes identified as significant have a variety of metabolisms/responses to environmental conditions.

Reviewer #2 (Comments for the Author):

- The "much easier" in the title is a bit awkward; consider revising (e.g. energy amendments shape the bacterial community structure in salinized farmland soil more than..."
- Line 18 - associated (add "d")
- Line 20 - phyla (plural); Delete "And" in the next sentence
- Line 23 - the phrase "drive the bacterial communities more stochastic" is a bit confusing and unclear; perhaps the form of "stochastic" needs to be changed; what properties of the bacterial communities are being driven? Is it structural changes? Shifts in abundance? Community assembly?
- Line 24 - the usage of "however" is inappropriate here
- Line 25 - the usage of the word "conducive" seems out of place; perhaps the authors mean "effective" or "influential"?
- Line 25 - the phrase "soil functional bacterial community species composition" - does this mean functional diversity or species diversity or relative abundance?
- Line 35 - is this "further" or "future"?
- Line 44 - revise "intensifies"
- Line 69 - is "infect" really what the authors mean? Or is it "affect"?
- Line 78 - "maintain" (remove "s"); "enable" (remove "s")
- Line 79 - "tolerate" (remove "s")
- Line 95 - what soil type?
- Manuscript may need language editing to correct some minor errors
- Randomization of pots? CRD? Were photosynthetic parameters measured for all leaves per plant?
- At what age and stage of the plant were the soil samples collected? This needs to be specified because bacterial community structure changes with the age of the plant. Processes that determine community assemblage also change with host plant age.
- Line 186 - is it 4 functions or just 3? C-cycle, energy, and N-cycle

- Line 191 - the plural of index is indices
- Line 365-369 - please check if the statement is correct (BA is not mentioned while DA is mentioned twice)
- Line 382 - "change in bacterial composition of soil had more effects on plant traits than the change of abundances" - this statement seems contradictory to the previous statement that increasing Proteobacteria abundance in RS under BA significantly enhanced PS
- What's the significance of the observed changes in the co-occurrence networks? How does this relate to the issue of soil salinization?
- What does the increased influence of stochastic processes on microbial community assemblage upon amendment of Biochar mean in relation to the problem of soil salinization? Will this effect be carried over until harvest? Or is the increased stochasticity observed only due to the short span of time since application of biochar?
- How does the study address the 3rd objective - To assess whether the change of soil bacterial community could affect the ability of host plants' salt resistance? How was host plant salt resistance evaluated? Is this through photosynthetic rate? Does high Ps rate automatically mean salt resistance?

Staff Comments:

Preparing Revision Guidelines

Please return the manuscript within 60 days; if you cannot complete the modification within this time period, please contact me. If you do not wish to modify the manuscript and prefer to submit it to another journal, please notify me of your decision immediately so that the manuscript may be formally withdrawn from consideration by Microbiology Spectrum.

Dear Editors and Reviewers:

Thank you very much for giving us an opportunity to revise our manuscript. We appreciate the editor and reviewers very much for their constructive comments and suggestions on our manuscript entitled "Energy amendments much easier shape the bacterial community structure in salinized farmland soil" (control no. Spectrum01012-22).

We have studied reviewers' comments carefully. Those comments are all valuable and very helpful for revising and improving our paper, as well as the important guiding significance to our researches. According to the reviewers' detailed suggestions, we have made a careful revision on the original manuscript. And we have thoroughly checked and polished the English of our manuscript with the help of a native English speaker John Wilmshurst. Revised portion are marked in red in the paper. We hope our revised manuscript can be accepted for publication. The main corrections in the paper and the responds to the reviewer's comments are as flowing:

Reviewer #1 (Comments for the Author):

Dear reviewer1,

I am very grateful to your comments for the manuscript. According with your advice, we amended the relevant part in manuscript. Revised portion are marked in red in the paper. And we have thoroughly checked and polished the English of our manuscript with the help of a native English speaker John Wilmshurst. Some of your questions were answered below:

Specific comments.

The title is uninformative as written since it's not clear what 'energy amendments' are.

Under your suggestion, we changed the title to: Carbon amendments shape the bacterial community structure in salinized farmland soil differ from non-carbon.

196, Unclear what is meant by 'pots were poured'? Does this mean 200ml of water added? What's the total volume of soil? What was the moisture content before and

after addition.

We apologize for the oversight and misunderstanding. 'pots were poured' should be 'Pots were irrigated with 200 ml distilled water'. We have now provided additional methodological details for the soil. The experimental soils were collected from an agricultural field cultivated with maize in the Aral District, Xinjiang Province, in northern China. The soil was air-dried, ground, and sifted through a 5mm sieve before the experiment (Tables S1). The experimental design included four treatments: 500g soil per pot (CK), 475g soil + 25g desulfurized gypsum addition per pot (DA), 485g soil + 15g biochar addition per pot (BA), and 460g soil + 15g biochar addition + 25g desulfurized gypsum addition per pot (DBA). Six replicates were maintained per treatment. We have added above information in the revised manuscript. Line 90-104.

l. 100. Biochar is simply described as maize, 400C, this is inadequate. Biochar itself is a complex compound, and the specific conditions under which it is produced can influence its properties, thus significantly more detail is needed here. Furthermore, the amounts of addition and % in soil are not given for any of the additions.

Thank you for your suggestion. We have added information about biochar in the revised manuscript and Supplementary Table 2. Line 97-100.

l104/5 What was the original salt concentration of these soils.

Thanks very much for your comment. Soil electrical conductivity (EC) is a parameter which represents the amount of salts in soils. The EC values of the original soils is 4.11 ± 0.06 dS/m. Relative content are reported in Supplementary Table S1.

P12. It's not clear what 'modified stochastic ratio' is or means, nor it is explained in the text. My general interpretation of the results is that there is a modest statistically significant impact of the biochar treatment on the microbial community, while other treatments were not significant. Thus, it's possible that the biochar may have, to some extent, selected for a different community, which would be determined by some, not

identified here, property that the microbes are responding to. It's not clear to me how this is 'stochastic'. Furthermore, without more experimental details of how amendment concentrations and how these effect total soil moisture and organic C contents, it's a bit difficult to fully interpret.

We thank for your comment and we sorry that we did not write this point clearly in the originally submitted manuscript. Deterministic assembly processes can result from either homogenous or variable selection with respect to environmental characteristics (Barnett et al. 2020). In this study, we considered that the process of soil bacterial community assembly should be deterministic in four treatments. But the MST values of the BA and DBA were significantly higher than CK and DA. Upon reviewing the literature, we found that soil organic carbon was the major drivers of bacterial β diversity and bacteria were more influenced by undominated-based stochastic processes (Chen et al. 2022). And the stochastic process of bacterial community assembly was influenced by soil carbon metabolism. Therefore, after adding the biochar with high organic carbon content, there is a transition in soil bacterial community assembly process from homogeneous selection to stochasticity. As your suggestion, we have revised the relevant content about MST and bacterial community assembly in the method and discussion in the revised manuscript. (Line 174–192 and Line 347-361)

lower part pg 14 & pg15, A number of studies are cited, however it's unclear how this work directly links to these studies and adds something new.

Thank you very much for your important suggestion. We have removed and altered some of the wording in the Discussion section about “Taxonomic composition of soil bacterial communities”. We have added some literatures about the effect of biochar addition on the relative abundance of bacterial species, and removed literature with low relevance. (Line 311-323 and Line 374-388)

P15 The last statement of the MS about effects of biochar is largely conjecture. This experiment did not directly test nutrient mobilization or microbial growth as a result

of biochar amendment to lend direct support to these ideas, since the microbes identified as significant have a variety of metabolisms/responses to environmental conditions.

Thank you for the above suggestion. We have removed the last statement and re-written the discussion section to include a more comprehensive view of the change of soil bacterial community could affect the ability of host plants' salt resistance. Line 398-427.

Thank you for reading our paper carefully and giving the above positive comments. We appreciate your warm work earnestly, and hope the correction will meet with approval. Once again, thank you very much for your comments and suggestions.

Reviewer #2 (Comments for the Author):

Dear reviewer2:

I am very grateful to your comments for the manuscript. According with your advice, we amended the relevant part in manuscript. According your advice, we have thoroughly checked and polished the English of our manuscript with the help of a native English speaker John Wilmshurst. Some of your questions were answered below:

- The "much easier" in the title is a bit awkward; consider revising (e.g. energy amendments shape the bacterial community structure in salinized farmland soil more than..."

According to your advice, we agree and changed the title to "Carbon amendments shape the bacterial community structure in salinized farmland soil differ from non-carbon"

- Line 18 -associated (add "d")

Thank for your suggestion, we have made it.

- Line 20 - phyla (plural); Delete "And" in the next sentence

We have modified it.

- Line 23 - the phrase "drive the bacterial communities more stochastic" is a bit confusing and unclear; perhaps the form of "stochastic" needs to be changed; what properties of the bacterial communities are being driven? Is it structural changes? Shifts in abundance? Community assembly?

Thank you for your suggestion. We revised the expression of this point as follow: amendments would relieve selection pressure and increase the stochasticity of community assembly of bacterial communities. Line 22-26.

- Line 24 - the usage of "however" is inappropriate here

We have modified it.

- Line 25 - the usage of the word "conductive" seems out of place; perhaps the authors mean "effective" or "influential"?

We have revised it according to your advice.

- Line 25 - the phrase "soil functional bacterial community species composition" - does this mean functional diversity or species diversity or relative abundance?

Yes, it means the relative abundance of soil functional bacterial species for "C-cycle", "Energy" and "N-cycle". We have modified it in the revised manuscript.

- Line 35 - is this "further" or "future"?

We are very sorry for this mistake. We have modified it.

- Line 44 - revise "intensifies"

Thank you for your suggestion. We have revised "intensifies".

- Line 69 - is "infect" really what the authors mean? Or is it "affect"?

We are very sorry for this mistake. We have modified it.

- Line 78 - "maintain" (remove "s"); "enable" (remove "s")

We have modified it.

- Line 79 - "tolerate" (remove "s")

We are very sorry for this mistake. We have modified it.

- Line 95 - what soil type?

Thank you for your advice. The soil type is classified as sandy loam. We have added this information in the revised manuscript.

- Manuscript may need language editing to correct some minor errors

According your advice, we have thoroughly checked and polished the English of our manuscript with the help of a native English speaker John Wilmshurst.

- Randomization of pots? CRD? Were photosynthetic parameters measured for all leaves per plant?

Thank you for pointing out this statement. To make it clear, we added the sentence in Line 109-113 to the following:” After 20 days of seedling growth (two leaves stage), three pots were randomly selected in each treatment to determine the photosynthesis and leaf water potential of seedlings and collect soil samples for microbial analysis. Measurements of photosynthesis were conducted using youngest fully expanded main-stem leaf (the 1st leaf from the apex).”

- At what age and stage of the plant were the soil samples collected? This needs to be specified because bacterial community structure changes with the age of the plant. Processes that determine community assemblage also change with host plant age.

Thank you for pointing to the missing information. In this study, we want to improve the farmland abandonment due to the soil salinity by use of amendments. Hence, the

experimental soils were collected from farmland abandonment where previous cultivated maize in the Aral District, Xinjiang Province, in northern China. We have added the above information in the revised manuscript. Line 90-94.

- Line 186 - is it 4 functions or just 3? C-cycle, energy, and N-cycle

We are very sorry for this mistake. We have modified it.

- Line 191 - the plural of index is indices

We are very sorry for this mistake. We have modified it.

- Line 365-369 - please check if the statement is correct (BA is not mentioned while DA is mentioned twice)

We are very sorry for this mistake. We have modified it in the revised manuscript.

- Line 382 - "change in bacterial composition of soil had more effects on plant traits than the change of abundances" - this statement seems contradictory to the previous statement that increasing Proteobacteria abundance in RS under BA significantly enhanced PS

Thank you for your suggestion. We have re-written this part in the revised manuscript (Line 397-427) to remove the ambiguity.

- What's the significance of the observed changes in the co-occurrence networks? How does this relate to the issue of soil salinization?

Thanks very much for your question. First, we found that the complexity of bacterial networks increased with the application of BA and DA, respectively. This means that the soil microbial communities were more resistant to salt stress relative to CK treatment. Second, different soil amendments could increase the interactions of different bacterial phyla. AND adding the biochar was able to enhance the interaction among the Proteobacteria phyla, while desulfurization was the Firmicutes phyla. Proteobacteria and Firmicutes were reported as "good indicators" for reflecting

changes of the main microbial groups and important resource for exploring halophilic enzymes and metabolic pathways for pollutant remediation in saline soil. We have re-written this part in the revised manuscript. Line 362- 388.

- What does the increased influence of stochastic processes on microbial community assemblage upon amendment of Biochar mean in relation to the problem of soil salinization? Will this effect be carried over until harvest? Or is the increased stochasticity observed only due to the short span of time since application of biochar?

Thank you very much for your good question. We generally considered that Soil properties among treatments in this study are broadly consistent, so the bacterial community in the soil should be the same development process. In our study, we found that stochastic processes dominated the soil bacterial community assembly only with biochar addition. Some studies have demonstrated increased resource supply can increase stochasticity under physicochemical conditions that do not impose strong selection (Dini-Andreote et al., 2015). Hence, we suppose that the increasing carbon-related bacterial species would enhance the function and role of soil microbial communities after adding the biochar, which may enhance the function and role of soil microbial communities to develop resistance to soil salinity stress. And we suggest that any addition of soil carbon amendment would increase the stochastic process of soil bacterial community assembly. We have added above sentences in the revised manuscript. Line 347-361.

- How does the study address the 3rd objective - To assess whether the change of soil bacterial community could affect the ability of host plants' salt resistance? How was host plant salt resistance evaluated? Is this through photosynthetic rate? Does high Ps rate automatically mean salt resistance?

Thank you for the above suggestion. We have re-written the discussion section to include a more comprehensive view of the change of soil bacterial community could affect the ability of host plants' salt resistance. Line 398-427.

Thank you for reading our paper carefully and giving the above positive comments. We appreciate your warm work earnestly, and hope the correction will meet with approval. Once again, thank you very much for your comments and suggestions.

November 29, 2022

Dr. Qisheng Han
Farmland Irrigation Research Institute
Road Honglidadao 380
Xinxiang, Henan
China

Re: Spectrum01012-22R1 (Carbon amendments shape the bacterial community structure in salinized farmland soil differ from non-carbon)

Dear Dr. Qisheng Han:

Thank you for submitting your revised manuscript to Microbiology Spectrum. I can now inform you that the Editorial Board has evaluated the manuscript Spectrum01012-22R1: Energy amendments much easier shape the bacterial community structure in salinized farmland soil.

The manuscript has been reviewed carefully by experts (comments given below). The Editor has advised that the manuscript will be considered for publication after minor revision.

Thank you for submitting your manuscript to Microbiology Spectrum. As you will see your paper is very close to acceptance. Please modify the manuscript along the lines I have recommended. As these revisions are quite minor, I expect that you should be able to turn in the revised paper in less than 30 days, if not sooner. If your manuscript was reviewed, you will find the reviewers' comments below.

When submitting the revised version of your paper, please provide (1) point-by-point responses to the issues raised by the reviewers as file type "Response to Reviewers," not in your cover letter, and (2) a PDF file that indicates the changes from the original submission (by highlighting or underlining the changes) as file type "Marked Up Manuscript - For Review Only". Please use this link to submit your revised manuscript. Detailed instructions on submitting your revised paper are below.

Link Not Available

Sincerely,

Wen-Li Chen

Reviewer comments:

Reviewer #2 (Comments for the Author):

Thank you very much; all my comments have been addressed satisfactorily. One last suggestion, though -- I suggest that the title be revised. Maybe you can consider the following: "Carbon amendments shape the bacterial community structure in salinized farmland soil differently compared to non-carbon treatment" or simply "Carbon amendments shape the bacterial community structure in salinized farmland soil"

Preparing Revision Guidelines

Please return the manuscript within 60 days; if you cannot complete the modification within this time period, please contact me. If you do not wish to modify the manuscript and prefer to submit it to another journal, please notify me of your decision immediately so that the manuscript may be formally withdrawn from consideration by Microbiology Spectrum.

Dear Editors and Reviewers:

Thank you very much for giving us an opportunity to revise our manuscript. We appreciate the editor and reviewers very much for their constructive comments and suggestions on our manuscript entitled "Carbon amendments shape the bacterial community structure in salinized farmland soil differ from non-carbon "(Spectrum01012-22R2).

We have studied reviewers' comments carefully. Those comments are all valuable and very helpful for revising and improving our paper, as well as the important guiding significance to our researches. According to the reviewers' detailed suggestions, we have made a careful revision in "Marked-Up Manuscript". Revised portion are marked in the manuscript (tracked changes). We hope our revised manuscript can be accepted for publication. The main corrections in the paper and the responds to the reviewer's comments are as flowing:

Dr. Qisheng Han

On behalf of all authors.

Reviewer #2

Dear reviewer2,

I am very grateful to your comments for the manuscript. According with your advice, we amended the relevant part in manuscript. Revised portion are marked in the manuscript (tracked changes). Here are responses to your comments:

1. Maybe you can consider the following: "Carbon amendments shape the bacterial community structure in salinized farmland soil differently compared to non-carbon treatment" or simply "Carbon amendments shape the bacterial community structure in salinized farmland soil"

We are grateful for the suggestion. We have revised the title to "Carbon amendments shape the bacterial community structure in salinized farmland soil".

Thank you for reading our paper carefully and giving the above positive comments. We appreciate your warm work earnestly, and hope the correction will meet with approval. Once again, thank you very much for your comments and suggestions.

December 15, 2022

Dr. Qisheng Han
Farmland Irrigation Research Institute
Road Honglidadao 380
Xinxiang, Henan
China

Re: Spectrum01012-22R2 (Carbon amendments shape the bacterial community structure in salinized farmland soil)

Dear Dr. Qisheng Han:

I am pleased to inform you that your manuscript has been accepted, and I am forwarding it to the ASM Journals Department for publication. You will be notified when your proofs are ready to be viewed.

Sincerely,

Wen-Li Chen
Editor, Microbiology Spectrum
